# People represent their own mental states more distinctly than those of others

Mark A. Thornton [1,2], Miriam E. Weaverdyck [3], Judith N. Mildner [1] & Diana I. Tamir[1,2]

One can never know the internal workings of another person—one can only infer others' mental states based on external cues. In contrast, each person has direct access to the contents of their own mind. Here, we test the hypothesis that this privileged access shapes the way people represent internal mental experiences, such that they represent their own mental states more distinctly than the states of others. Across four studies, participants considered their own and others' mental states; analyses measured the distinctiveness of mental state representations. Two fMRI studies used representational similarity analyses to demonstrate that the social brain manifests more distinct activity patterns when thinking about one's own states vs. others'. Two behavioral studies complement these findings, and demonstrate that people differentiate between states less as social distance increases. Together, these results suggest that we represent our own mind with greater granularity than the minds of others.

[1] Department of Psychology, Peretsman Scully Hall, Princeton University, Princeton, NJ 08540, USA. [2] Princeton Neuroscience Institute, Princeton University, Princeton, NJ 08540, USA. [3] Department of Psychology, 1285 Franz Hall, University of California, Los Angeles, Los Angeles 90095 CA, USA. Correspondence and requests for materials should be addressed to M.A.T. (email: mark.allen.thornton@gmail.com)

There exists an impassible divide between self and other. On one side lives one's own mind—directly and vividly accessible[1–3]. On the other side live the minds of others, opaque and inaccessible. To know the mental states of others we must piece together fragile inferences from perceptible clues, such as actions, facial expressions, voice tone, and context[4–6]. Even the richest menu of sensory information cannot compare to our own first-hand experiences: we can see a friendly smile or hear them yell, but we cannot feel the soreness in their muscles or pounding of their heart. The sheer quantity of experience with ourselves which we each accumulate over time makes the self qualitatively different from others[7–9]. How does this direct, enriched, and plentiful access to one's own mind affect the way we think about the contents of our own minds vs. the minds of others?

Here we test the hypothesis that people represent their own mental states with greater granularity than the mental states of other people. This prediction follows naturally from the premise that people know more about their own minds than those of others: with greater insight comes the potential for greater granularity. Thus, when thinking about one's own mind, a person might make nuanced distinctions between their own thoughts and feelings. For instance, one might observe the fine distinctions between one's own melancholy, despair, and ennui, or the clear distinction between one's sleepiness and laziness. In contrast, when people consider the minds of others, they might instead lump such states together.

This hypothesis reflects an intra-mind analog of the inter-mind outgroup homogeneity effect[10]. In the established outgroup homogeneity effect, people in the outgroup are viewed as more similar to each other than members of the ingroup; in the hypothesized other minds homogeneity effect, the states within the mind of another person are viewed as more similar to each other than the states within one's own mind. A rich literature supports the notion that we represent the minds of certain others as impoverished. For instance, perceivers routinely dehumanize members of outgroups by imputing less capacity for mental agency and experience to them[11,12]. Here we suggest that, relative to one's own mind, we may represent the minds of all others as impoverished.

To test this hypothesis, participants in four studies considered their own mental states and the mental states of other people at varying social distances. We used both functional magnetic resonance imaging and behavioral approaches to measure the distinctiveness of mental state representations. In Studies 1 and 2, we measured the distinctiveness of neural representations of states, operationalized as patterns of neural activity. Specifically, we used representational similarity analysis[13,14] to measure the similarity between neural activity patterns for each pair of mental states examined. We predicted that we would observe more distinctive activity patterns when participants thought about their own states, and more similar activity patterns when they thought about others' states. Moreover, we predicted that this effect would occur within the social brain network, a set of brain regions sensitive to mental state representation[15] that includes the medial prefrontal cortex, medial parietal cortex, the temporoparietal junction, and the anterior temporal lobe.

In Studies 3 and 4, participants made explicit ratings of the similarity between mental states for multiple target individuals. This behavioral data offers a convergent measure of the same hypothesis: we predicted that participants would rate their own states as less similar to each other than the states of a given other. Additionally, in Studies 2–4, we examined whether social distance modulates the extent to which people differentiate between others' states. In particular, we test whether people differentiate between the states of close others more so than the states of distant others.

Across these four studies, evidence supports the hypothesis that people represent their own mental states more distinctly than those of others. In Study 1, neural patterns are more distinct for one's own states than for those of a socially distant other. We observe the same result in Study 2, and also observe that one's own states are represented more distinctly than those of a close other. Studies 3 and 4 provide behavioral replications of the fMRI results from Studies 1 and 2. People rate their own states as more distinct from each other than they rate others' states. Ratings of state similarity increase with increasing social distance from the self. Together these findings support the conclusion that people think about their own minds with greater granularity than the minds of others.

## Results

**Study 1.** In Study 1 we tested whether neural representations of one's own mental states are more distinctive than representations of a socially distant other's mental states. By examining a target person very dissimilar to the self, we aimed to maximize any observable difference in the granularity of mental state representation between self and other. On each trial of the experiment, participants rated which of two scenarios—presented as visual images—would better elicit one of 30 mental states in either themselves or the socially distant (far) target (Fig. 1). Consistent with our prediction, results of representational similarity analysis indicated that participants represented their own mental states more distinctively than the far target's states. Searchlight analysis indicated that pattern similarity between mental states was significantly greater for the far target than for the self in the medial prefrontal cortex, portions of the dorsolateral prefrontal cortex, and the left temporoparietal junction (Fig. 2; Table 1). This effect also manifested in a t-test on pattern similarity within the independently defined social brain network (mean correlation difference = 0.016, $d = 0.46$, $p = 0.017$).

**Study 2.** Study 1 supported the hypothesis that people represent their own mental states more distinctly than the states of others. However, in that study, we compared the self to a socially distant target person. This raises the question of whether people represent the self with more granularity than all other social targets, or just socially distant targets. To answer this question, we conducted a second fMRI study of similar design. In this study, we presented participants with three target people: the self and the far target, as in Study 1, but also a socially proximal target, such as a close friend or family member. If participants represent their own states more distinctly than even those of the close target, this would suggest that they represent their own minds with a uniquely high level of granularity. We also varied low-level aspects of the paradigm (Fig. 1), including the set of mental states, the stimulus modality (text instead of images), and the timing and blocking of trials, to ensure that our conclusions generalize across these design feature.

An omnibus ANOVA across self, close, and far targets indicated that average neural pattern similarity changed significantly within the dorsal and ventral medial prefrontal cortex, the precuneus and posterior cingulate, bilateral temporoparietal junction, the anterior temporal lobe, and dorsolateral prefrontal cortex (Fig. 3a; Table 1). As predicted, pairwise t-tests revealed that state-specific activity patterns were more distinct for the self than the close target within medial prefrontal cortex, the anterior temporal lobe, and precuneus (Fig. 3b) and for the self than the far target states within all the regions detected in the omnibus results, except the anterior temporal lobe (Fig. 3c). The pairwise comparison between close and far others did not yield any statistically significant results. No brain regions showed the

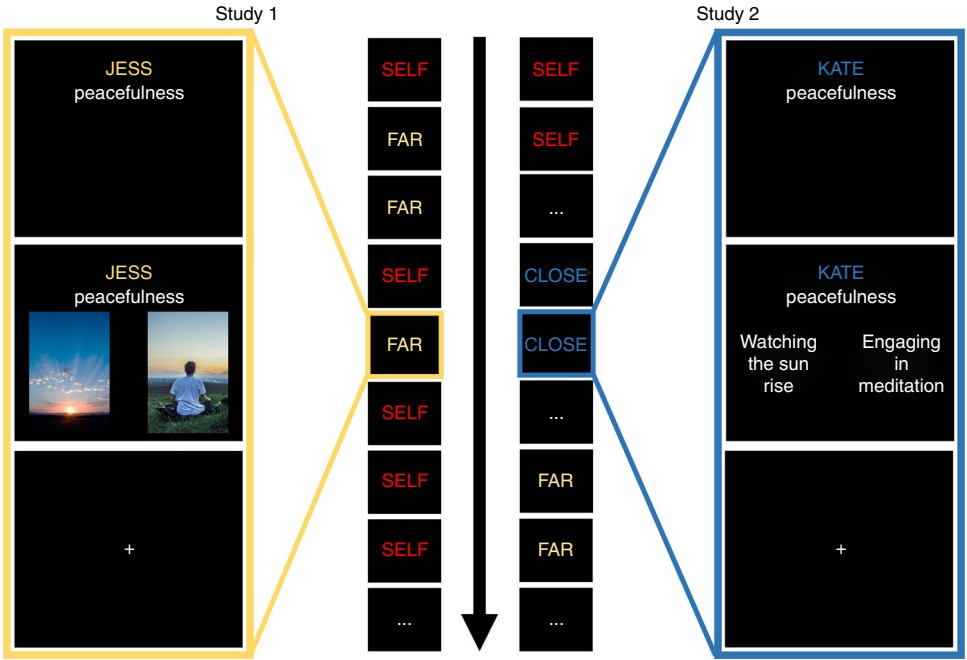

**Fig. 1** Experimental paradigm schematic for Studies 1 and 2. Participants in Studies 1 and 2 judged which of two scenarios would elicit more of a given mental state in a given target person. Scenarios were presented via images in Study 1, and via text in Study 2. In Study 1, trials about the self and far target were randomly intermixed. In Study 2, trials were grouped into target blocks (self, close, and far) within each run

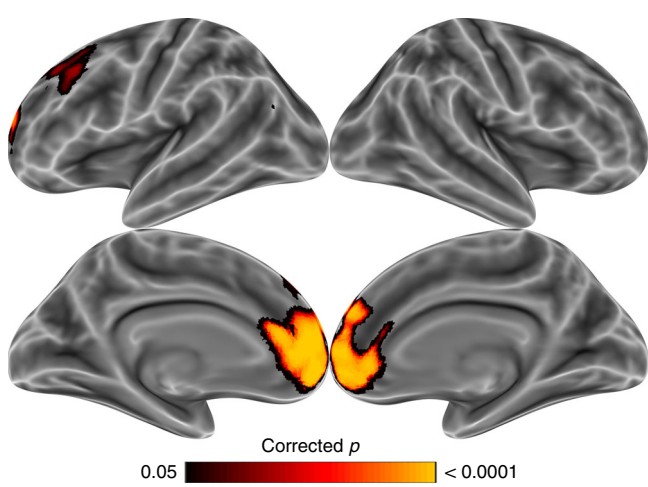

**Fig. 2** States distinctiveness is greater for self than for other (Study 1). A pairwise $t$-test on the average similarity of mental state-specific activity patterns revealed that state distinctiveness was greater for the self than for a socially distant other within regions implicated in social cognition, including the medial prefrontal cortex, the temporoparietal junction, and portions of the dorsolateral prefrontal cortex. The family-wise error rate was controlled ($p < 0.05$) via permutation testing with TFCE

Corrected $p$

0.05 — < 0.0001

reverse pattern of results, where states were represented more distinctly for others than for the self. In addition to the searchlight analysis, we interrogated patterns of activity within the significant regions in Study 1 (Fig. 2) using pairwise $t$-tests. Within these regions, state-specific activity patterns were more similar for the close target than the self ($\Delta r = 0.004$, $d = 0.64$, $p < 0.0006$), and more similar for the far target than the self ($\Delta r = 0.05$, $d = 0.82$, $p < 0.00003$). Activity patterns were not significantly more similar for the far than the close target ($\Delta r = 0.006$,

$d = 0.10$, $p > 0.57$). These results thus support the hypothesis that we represent our own minds with uniquely high granularity, outstripping even the closest social targets.

**Study 3**. Studies 1 and 2 established that the brain represents other people's mental states less distinctly than one's own. Does this implicit neural effect generalize to explicit behavior? In Study 3, we examined whether there was convergent behavioral evidence to support the findings of the fMRI experiments. If so, this would strengthen the conclusion suggested by the neural data, and also open up a more economical approach to studying this phenomenon. In this experiment, participants rated the pairwise similarity between the mental states within the minds of each of three target people: themselves, and the close and far targets from Study 2.

Results replicated Studies 1 and 2, such that people judged their own mental states to be less similar to each other than the states of others (Fig. 4 and 5c): mixed effects modeling the overall effect of self vs. other on state-similarity rating was statistically significant in the predicted direction ($b = 0.15$, $\beta = 0.04$, $t$ (46.19) = 2.194, $p = 0.033$), with marginal $R^2 = 0.002$ and conditional $R^2 = 0.40$. Pairwise differences were significant between the self and far target ($b = 0.18$, $t(48.1) = 2.11$, $p = 0.040$) but not the self and close target ($b = 0.13$, $t(45.1) = 1.37$, $p = 0.18$) or the close and far targets ($b = 0.05$, $t(47.0) = 0.49$, $p = 0.63$).

**Study 4**. Studies 1–3 provide evidence that people represent their own states more distinctly than they represent the states of socially close or far others. Does this effect reflect only the self-other divide, or can it be extrapolated to social distance more generally? The results of Studies 2–3 do not resolve this question: the means of the close and far target conditions suggest that people may represent the states of close targets with greater granularity than the states of far targets, but in neither study is the close-far distinction itself statistically significant. Study 4 was designed to resolve this question by testing whether the self-other

**Table 1 Regions showing effects of social distance on neural pattern dissimilarity**

| Contrast/Region Name | x | y | z | Minimum $p_{corr}$ | Voxel extent |
|---|---|---|---|---|---|
| *Self > Far t-test (Study 1)* | | | | | |
| Medial prefrontal cortex | −8 | 48 | −4 | 0.0001 | 4509 |
| L Dorsolateral prefrontal cortex | −28 | 22 | 40 | 0.0315 | 386 |
| Dorsal medial prefrontal cortex | −10 | 42 | 42 | 0.0342 | 98 |
| L Temporoparietal junction | −48 | −70 | 24 | 0.0481 | 5 |
| *Omnibus ANOVA (Study 2)* | | | | | |
| Precuneus/Posterior cingulate | 10 | −52 | 16 | 0.0000 | 3119 |
| R Temporoparietal junction | 48 | −68 | 24 | 0.0000 | 1431 |
| Medial prefrontal cortex | 0 | 32 | −12 | 0.0003 | 2768 |
| L Temporoparietal junction | −42 | −72 | 32 | 0.0058 | 655 |
| R Dorsolateral prefrontal cortex | 22 | 24 | 42 | 0.0081 | 575 |
| R Anterior temporal lobe | 54 | −10 | −28 | 0.0132 | 141 |
| L Dorsolateral prefrontal cortex | −24 | 12 | 50 | 0.0476 | 8 |
| *Self > Close t-test (Study 2)* | | | | | |
| Posterior cingulate | −6 | −52 | 28 | 0.0039 | 1432 |
| R Anterior temporal lobe | 54 | −6 | −30 | 0.0155 | 581 |
| Ventral medial prefrontal cortex | −2 | 28 | −10 | 0.0225 | 1116 |
| Medial prefrontal cortex | −2 | 48 | 20 | 0.0319 | 334 |
| *Self > Far t-test (Study 2)* | | | | | |
| Medial/lateral prefrontal cortex | −2 | 26 | −20 | 0.0000 | 8204 |
| Precuneus/Posterior cingulate | 10 | −52 | 14 | 0.0000 | 4661 |
| R Temporoparietal junction | 48 | −66 | 20 | 0.0000 | 2891 |
| L Temporoparietal junction | −44 | −76 | 32 | 0.0043 | 1153 |

Coordinates refer to the Montreal Neurological Institute stereotaxic space. Voxels are statistically significant ($p < 0.05$, corrected) accounting for multiple comparisons. Italicized titles indicate the four statistical contrasts

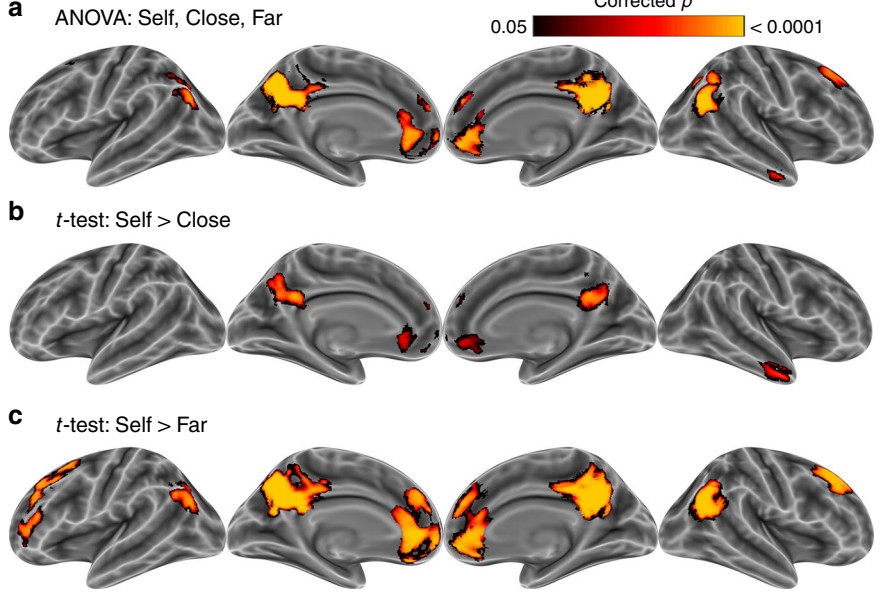

**Fig. 3** State distinctiveness is greater for self than for other (Study 2). **a** A three-way ANOVA (target: self, close, far) revealed that state distinctiveness changed as a function of target within many regions implicated in social cognition, including the medial prefrontal cortex, the temporoparietal junction, and precuneus/posterior cingulate. Pairwise *t*-tests indicated that the states were **b** less distinct for close others compared to the self, and **c** less distinct for far others compared to the self. No regions showed the reverse pattern of more distinct patterns for more distant targets. Family-wise error rate was controlled ($p < 0.05$) via permutation TFCE

difference generalizes to people who are more vs. less similar to the self. This study repeated the similarity rating task in Study 3, but focused on a smaller number of states within just the close and far targets.

People rated a close target's states with more granularity than a far target's states: In mixed effects models, we observed greater rated similarity between the far target's states than between the close target's ($b = 0.23$, $\beta = 0.07$, $t(134.2) = 4.54$, $p = 0.000013$, marginal $R^2 = 0.005$, conditional $R^2 = 0.39$; Fig. 5d). The results of a continuous version of this analysis, which treated social distance as composite of similarity, familiarity, and closeness ratings, converged with the categorical model based on target ($b = .13$, $\beta = .08$, $t(168.45) = 4.37$, $p = 0.000022$, marginal $R^2 = 0.007$, conditional $R^2 = 0.40$). Together these results suggest that

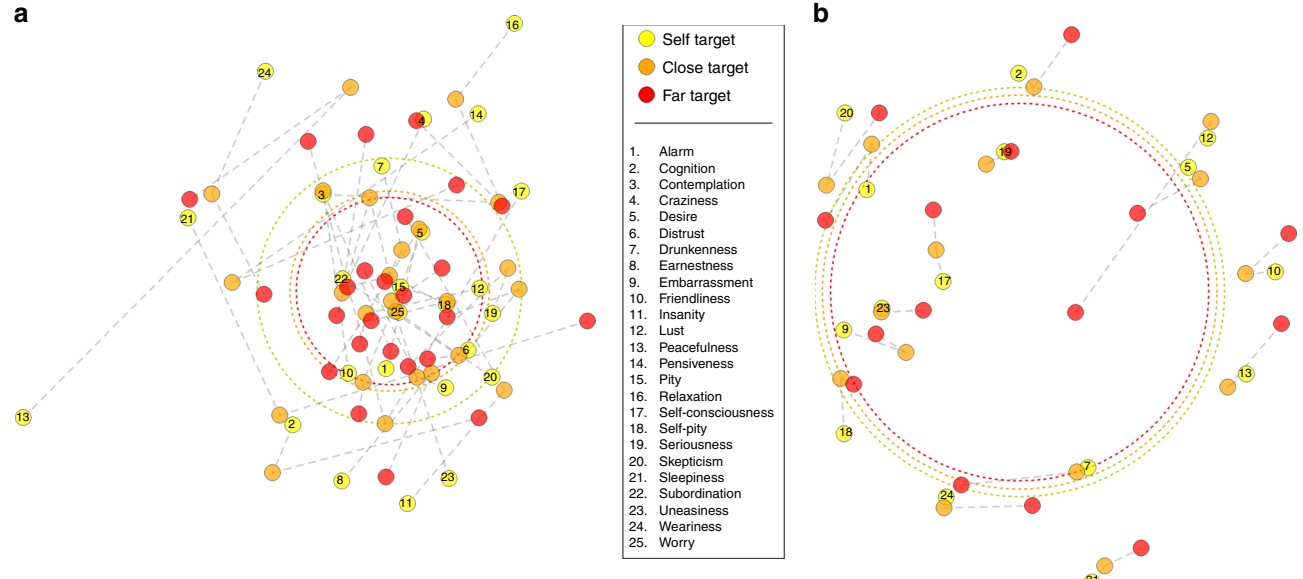

**Fig. 4** Multidimensional scaling of state similarity in experiments with three targets. The representational space of mental states is larger for the self (yellow) than close (orange) or far (red) targets. Each point represents one mental state; dashed gray lines connect the same state across targets. The distances between points reflect **a** the dissimilarities between state patterns in Study 2 in the regions which showed significant effects in the Study 1 searchlight, and **b** explicit dissimilarity ratings from Study 3. The dotted circles reflect the average pattern dissimilarity for each target. Scaling was conducted independently for each target, with results rotated into similar orientations within-study to facilitate comparison. Note: while closer states are generally more similar to each other, distances on this 2-D manifold do not perfectly reproduce the neural or rated similarity. The generation of this figure is described in the Supplementary Methods (p. 4–5)

the self-other difference in the granularity of mental state representation may extend to social distance more generally.

## Discussion

People represent their own mental states more richly than they represent the mental states of others. Two neuroimaging studies demonstrate that the social brain represents one's own states with greater granularity than the states of distant others, and even close friends and family. Study 3 provides convergent behavioral evidence for this effect, demonstrating that people explicitly rate their own mental states as more distinct than the states of others. Together, these results indicate that the unique access we have into our own minds allows us to represent our mental states with a degree of clarity and richness unmatched when considering the minds of others.

Further, the results of Study 4 indicate that this self-other difference in granularity may reflect a more general effect of social distance: people rated the states of a socially distant target as more similar to each other than the states of a socially proximal target. This suggests that the distinctiveness of mental state representation may decrease as the social distance of the target increases. Under this interpretation, the self would effectively be an example of an extremely—indeed, maximally—socially proximal target. If this social distance extension is correct, then it could help explain a number of existing findings in the psychological literature, such as why people fail to perceive the richness of outgroup members' minds[11,16,17]. Just as people often fail to perceive the differences between socially distant individuals[10], here we show that they also fail to perceive differences within individuals, from state to state. Such outgroup biases can have negative consequences, including explicit bias and implicit insensitivity to others' pain[18,19]. Previous attempts to attenuate outgroup bias that have focused on individuating outgroup members have successfully reduced intergroup discrimination[20]. The current findings hint that individuating the states within

outgroup individuals' minds may further diminish the gap between us and them, although this hypothesis requires direct testing. That said, Study 4 tested only a subset of mental states, chosen specifically to maximize our power to detect differences between close and far others. As such, it is not yet known whether this effect of social distance generalizes to the full set of possible mental states, nor to the full set of possible social targets.

Why do people represent their own states with greater granularity than those of others? One possibility is that people simply have more information about themselves. For example, people have privileged access to the content of their own minds, physiological access to signals from their bodies. In addition to these qualitative differences, the self is also quantitatively different, as people also have uniquely extensive experience with themselves, in comparison with even the closest other. People may also care more about their own states than those of others. If so, then effort might explain the observed difference in granularity. Future research may help to clarify the relative contributions of these factors. For example, studies could give participants varying degrees of insight into a social target's internal states; or, as in Study 4, studies could use social distance to manipulate familiarity and motivation. Given that we observe an effect of distance on the distinctiveness of mental state representations, these factors may be particularly promising candidates for future study. Alternatively, one could attempt to mimic the unique information channels available to the self by giving pseudo-introspective access into another person's mind (i.e., telling them everything someone is thinking) to see if such access increased the distinctiveness of state representations.

Some mental states show precipitous drops in distinctiveness between self and other, whereas other states show more gradual drops. In particular, it appears that very low-energy states, such as peacefulness, relaxation, weariness, and sleepiness show the greatest drop in neural granularity from self to other (Fig. 3a). Previous research suggest that these states may be defined by their low social impact[15]. That is, these states may not be particularly

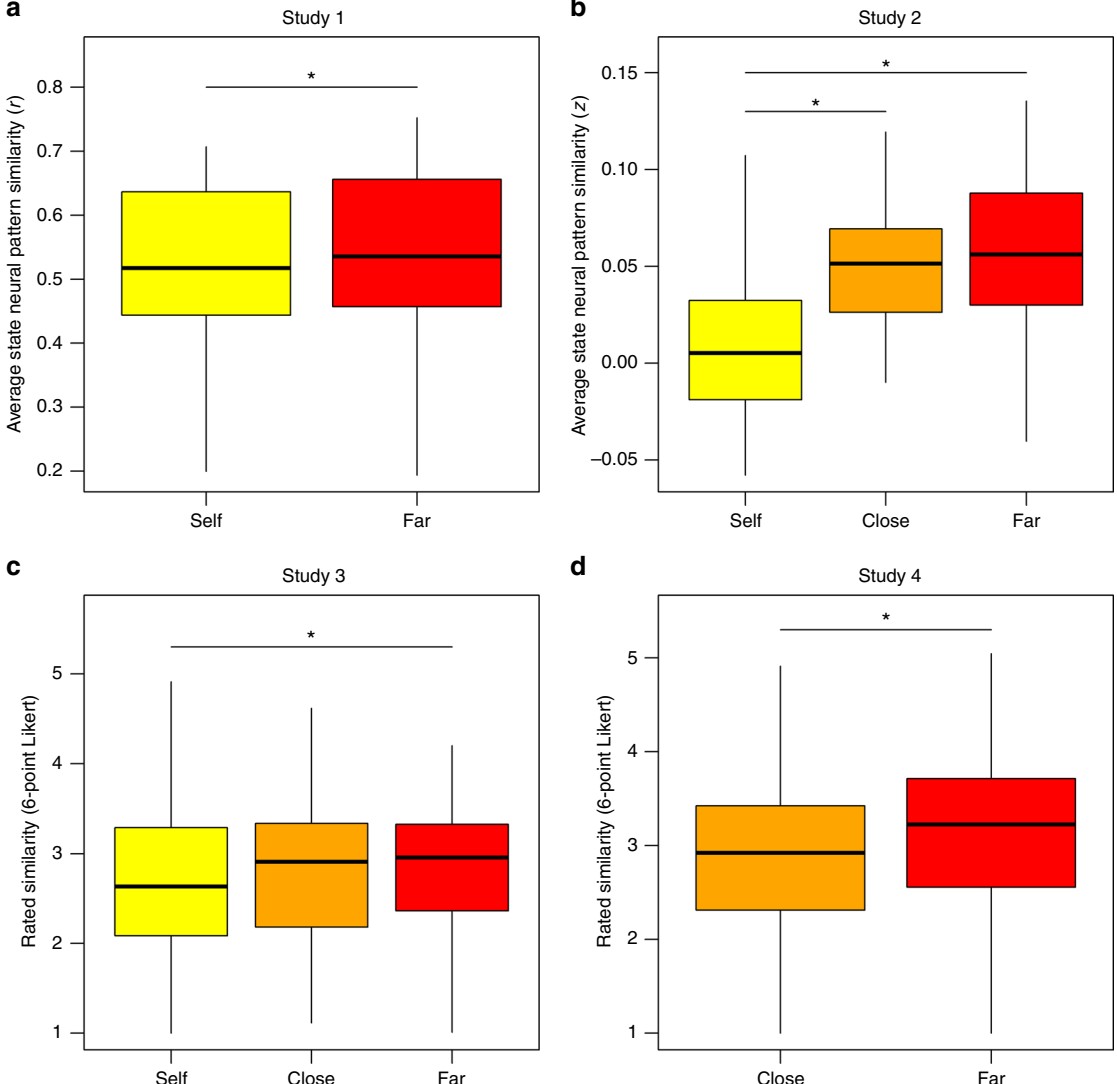

**Fig. 5** Effects of self vs. other on state similarity by target. Box plots represent the distributions of average similarity between states within self, close, and far target people. **a** In Study 1, and **b** Study 2, state similarity was measured via correlating patterns of brain activity. **c** In Study 3, and **d** Study 4, state similarity was measured by explicit behavioral ratings. Centerline indicates median, boxes indicate interquartile range, and whiskers indicate 1.5 times the interquartile range. Asterisks denote statistically significant differences between pairs of conditions ($p < 0.05$), as measured by $t$-tests (Studies 1 and 2) or mixed effects models (Studies 3 and 4)

relevant to social interactions. If so, this may explain why the brain does not bother to represent them with great granularity, specifically in the minds of others.

The present findings may help to explain why judgments of others' minds tend to reflect an egocentric bias towards one's own thoughts, feelings, and preferences[21,22]. Specifically, the richness of one's own mind may lead people to use it as a model for others'[9,23]. Even if we do not explicitly simulate others' minds[24], the conceptual richness of our own mental states may exert a gravitational influence which bends our theory of mind in an egocentric direction. In other words, the relation between one's own states and another's states might be analogous to the relation between platonic objects and real objects: the former serving as ideal versions of the latter, shaping the way we think about them. That said, rich representations are not necessarily accurate ones. People have notable lapses in their ability to truly know their own state, traits, and actions[25–27]. The richness of their representations of self may serve to enhance the illusion that we do truly know ourselves.

The present findings may also have implications for how we understand social prediction. People use knowledge about current states to make predictions about future states[28]. For example, if I know I am currently hungry, I can predict that I will soon be angry, lest I eat some food. Recent work suggests that if one state predicts another state – as hunger predicts anger – then the two states will elicit similar patterns of brain activity[29]. Here we find that the others' mental states generally elicit more similar patterns of brain activity than one's own. If this heightened similarity translates into heightened transitional probabilities, it would suggest that other people are less predictable than oneself, as others are more likely to transition from any one state to any other. Indeed, it seems plausible that people have a better predictive model of their own minds than those of others. This interpretation aligns with our predictive coding account of social cognition[30], which suggests that the way the brain organizes social information allows people to make predictions about the social future.

The results of this investigation shed new light on the well-established effects of thinking about self and other on univariate levels of brain activity[31–35]. Here, we observe univariate results which are qualitatively quite similar to those shown in previous studies. (Supplementary Fig. 1). Specifically, thinking about one's own mental states elicited substantially greater activity in the medial prefrontal cortex than did thinking about the mental states of others. These same univariate contrasts also replicate a common effect finding greater activation for other vs. self in most of the rest of the social brain network—including medial parietal cortex, temporoparietal junction, the anterior temporal, and (other) portions of medial prefrontal cortex.

Interestingly, even though medial prefrontal cortex may increase its univariate activity during self-referential thought while medial parietal cortex decreases its activity, both regions demonstrate the same multivariate similarity pattern. That is, both regions show greater differentiation of self-state patterns than other-state patterns despite showing opposite univariate responses to self and other. This discrepancy calls into question that assumption that greater activity in the medial prefrontal cortex is a unique signature of enhanced self-referential processing. Instead, the present results suggest that the brain may exert different univariate levers to achieve the same representational changes. These results thus emphasize the importance of considering not just overall activity levels within voxels, but also the multivoxel patterns that they form.

In conclusion, the present results demonstrate that people represent their own states with a granularity unmatched when they think about the minds of others. This self-other distinction may influence the process by which we take others' perspectives, and determine how successfully we can predict their future mental states. Understanding how the distinctiveness of mental state representations varies across different social roles and relationships, and along different psychological dimensions, may reveal much about how we tailor our theory of mind to fit the people in our lives, including ourselves.

## Methods

**General methods**. We report all data exclusions, manipulations, and measures. Sample sizes were determined a priori via resampling-based or parametric power analyses (see Supplementary Methods, p. 1). Statistical tests were two-sided. All participants provided informed consent in a manner approved by the Princeton University Institutional Review Board. We complied with all relevant ethical regulations on working with human subjects.

**Study 1**. Participants ($N = 30$; 14 female, 15 male, 1 nonbinary; mean age 20, age range 19–27) were recruited via the Princeton University Credit and Paid Study Pools. Prior to data analysis, two participants out of 32 recruited were excluded due to low response rates (<50%) in the behavioral paradigm. Participants in both neuroimaging studies (Studies 1 and 2) were right-handed or ambidextrous, native English speakers, reported no history of neurological problems, had normal or corrected-to-normal vision, and were screened for standard MRI exclusion criteria (e.g., metal implants, pregnancy).

Participants underwent functional neuroimaging while considering the mental states of two target people (Fig. 1): the self, and a socially distant (far) target constructed by the experimenters to be dissimilar from participants based on their self-reported politics, religion, hobbies, and college major. At the onset of each trial, a mental state term (e.g., awe, friendliness, uneasiness, etc.) and a target name (e.g., self) were presented for 0.75 s. These words remained on screen while two images (e.g., for awe: a picture of whales or a picture of an aurora) appeared for 3.45 s on the screen below the mental state term and target, one on the left side of the screen, the other on the right side. Participants decided which of the two images they thought would be more likely to elicit the mental state in the target person, and indicated their response using a button box in their left hand by pressing either the middle or index finger buttons for the left and right scenarios, respectively. Jitter was added stochastically between trials: range 0–4.2 s, in 1.4 s increments, randomly selected from a Poisson distribution with a mean of 1.4 s.

Participants saw the two targets (self and far) paired with each of 30 mental states once in each of 12 runs. For each target, participants saw each of 12 pictures twice over the course of the experiment; every trial within a target showed a unique pair of these pictures. Self and far target trials were randomly intermixed within

each run. To acquaint participants with the task, they completed a brief practice session prior to entering the fMRI scanner. The imaging studies were presented via PsychoPy[36] in Python 2.7 (https://www.python.org/).

Functional MRI data were acquired at Princeton University using a 3T Siemens Prisma scanner with a 64-channel head coil. T2*-weighted images were collected during the mental state judgment task (gradient echo multi-slice EPI: 2 mm isotropic voxels, repetition time [TR] = 1.4 s, echo time [TE] = 32 ms, flip angle [FA] = 70°). Twelve functional runs of 209 TRs each were collected from each participant over the course of the experiment. We also collected a T1-weighted high-resolution anatomical scan for each participant (MPRAGE: 1 mm isotropic voxels, TR = 2.3 s, TE = 2.27 ms, FA = 8°). All MRI data were preprocessed using standard procedures, with FSL[37] for motion correction, slice timing correction, and unwarping, and DARTEL[38] for coregistration of functional and structural volumes, and normalization to MNI space. Following preprocessing, the data were entered into a GLM consisting of boxcar regressors for each target-state combination, convolved with a canonical hemodynamic response function. Additional covariates of no interest were included to control for run means and trends, and head motion parameters. General linear models (GLMs) were run using SPM12 (Wellcome Department of Cognitive Neurology) with the SPM12w extensions (https://github.com/wagner-lab/spm12w) to prepare fMRI data for pattern analysis.

To test the hypothesis that people represent their own mental states more distinctly than those of others, we conducted representational similarity analysis (RSA) on the regression coefficient patterns from the GLMs[13]. To remain agnostic with respect to the spatial scale at which relevant neural patterns manifest and change, we conducted RSA at two spatial levels, as in our previous neuroimaging work:[15] using a searchlight to examine local patterns[14] and an independently-selected set of brain regions to examine broadly distributed patterns.

In a searchlight RSA analysis, state-specific GLM contrasts for each target person were extracted from within a small, approximately spherical volume with a 4 voxel (~9 mm) radius centered at each voxel in the brain in turn. The Pearson correlation distance between each pair of patterns was calculated to produce neural dissimilarity matrices for each searchlight position. These matrices ($60 \times 60$) represented the neural dissimilarity between each pair of mental states for each target person. Within each target person, the correlations between each pair of mental states were averaged to produce a single estimate of the average pattern dissimilarity for that target. Only the lower triangular portions of the pattern dissimilarity matrices were used, since the diagonal of a correlation matrix is uninformative, and the upper triangular portion is redundant with the lower. These average pattern similarity maps were then spatially smoothed with a 6 mm FWHM Gaussian kernel and entered into a paired t-test (self vs far). The family-wise error rate across voxels was controlled via maximal statistic permutation testing with threshold-free cluster enhancement (TFCE)[39].

In the independently-selected social brain analysis, the same representational similarity analysis was repeated using a single set of voxels instead of a moving searchlight volume. The selected voxels were sensitive to mental state content, defined by showing a significant effect of state (voxelwise $p < 0.0001$) in an omnibus ANOVA in a previous study of similar design[15]. Selected regions included medial prefrontal and parietal cortices, the temporoparietal junction, superior temporal sulcus and anterior temporal lobe, as well as portions of the lateral prefrontal cortex. Contrast patterns across the entire set of selected voxels were correlated to estimate the neural similarity between mental states. These correlations were then averaged within target person to produce estimates of the pattern distinctness of each target, just as in the searchlight analysis. These averages were compared via paired t-tests across participants, as in the searchlight analysis.

**Study 2**. Participants in Study 2 ($N = 35$; 23 female, 12 male; mean age 21, age range 18–31) were recruited from the same pool as Study 1. Prior to data analysis, four participants were excluded from an original sample of 39 due to excessive movement within the scanner (i.e., in the majority of runs, moving more than 2 mm, or more than 0.5 mm more than 5 times).

Participants considered the mental states of three target people (Fig. 1): the self and far target, as in Study 1, as well as a close target, nominated by the participant as someone to whom they felt close (e.g., a friend or relative) and similar. This target was of intermediate social distance, between the self and the far target. The task that participants engaged in was the same as that in Study 1, with the following exceptions. First, the timing of the trial was different, with 0.5 s for the initial target/state prompt, 4 s scenario presentation plus response, and a slightly different jitter distribution (range 0–9 s, in 2.25 s increments, randomly selected from a Poisson distribution with a mean of 1.53 s). Second, instead of images, participants judged which of two text-based scenarios would better elicit each mental state in the target (e.g., for friendliness: calling someone just to talk or picking up a neighbor from the airport). Third, trials for each target person were grouped in blocks rather than intermixed, with one block for each target in each run. The order of trials within blocks and blocks within runs were randomized for each participant. Finally, a different set of 25 mental states were presented.

Imaging acquisition and parameters were the same as in Study 1, except for the TR (2.25 s) and run length (243 TRs). Preprocessing and GLM were also conducted in the same manner for both imaging studies. The searchlight and social brain network RSAs were also conducted in the same manner, allowing for the differences in target and state numbers (i.e., they produced $75 \times 75$ neural

dissimilarity matrices resulting from 3 targets and 25 states). The only difference was that these dissimilarity matrices were controlled for partial correlations in the design matrix induced by the block design (see Supplementary Methods, p. 4).

Statistical procedures differed slightly from Study 1. A voxelwise one-way repeated measures ANOVA with three levels (targets: self, close, far) was conducted first to test for any region showing a difference in mental state distinctiveness across targets. This was supplemented by subsidiary pairwise paired t-tests comparing the average pattern distinctiveness (dissimilarity) between each pair of targets. The family-wise error rate of each statistical map was controlled via maximal statistic permutation testing with TFCE. The Bonferroni correction was used to adjust for multiple comparisons with respect to the three unique t-tests comparing each pair of target people. Thus, results were controlled for multiple comparisons both across voxels, and across pairs of target people.

The network-level RSA was conducted similarly to Study 1. However, instead of analyzing the entire social brain network, we analyzed patterns from within the regions which showed a significant effect in Study 1. Thus, these regions were not just sensitive to mental state representation in general, but to the self-other effect on pattern similarity in particular. The inference was conducted via three pairwise one-sample t-tests on the changes between different target people in average state-specific pattern correlation (i.e., the distinctiveness of mental state representations).

**Study 3**. Participants in the online behavioral experiments (Studies 3 and 4) were recruited via Amazon Mechanical Turk, using TurkPrime[40] and then directed to Qualtrics-based surveys. Participants in Study 3 ($N = 46$; 22 female, 24 male; age range: 18–55, specific numeric ages not collected) engaged in a mental state-similarity rating task. Before beginning the task, they provided information to construct artificial biographies for the far target, as in Studies 1 and 2. They also nominated a close target in the same way as in Study 2. On each trial of the task, participants saw the names of one target and two mental states (e.g., self with embarrassment and sleepiness). They rated how similar the experiences of the two mental states were for that target on a 6-point Likert-type scale. Across the task they rated all unique pairs of 15 mental states for each of the three targets, for a total of 315 trials. Ratings were grouped by target person, and the orders of target blocks and of mental state pairs within each block were randomized for each participant. The 15 rated states were those that showed the greatest reduction in distinctiveness as a function of psychological distance in Study 2. Participants also rated how similar, close, and familiar the distant target person was to themselves.

Responses were analyzed using linear mixed effects modeling with the lme4 package in R. Maximal models were fit to predict participants' judgements of the pairwise similarity between mental states, with a fixed effect of target, random intercepts of participant and pair of states, and random slopes for target within participant and pair of states. The fixed effect of target was expressed using a binary contrast distinguishing the self from the other two targets. The statistical significance of fixed effects parameters was calculated via the Satterthwaite approximation for degrees of freedom. Least square means post-hoc tests were used to compare each of the three targets to one another, with the fixed effect of target dummy coded in this instance.

**Study 4**. In Study 4, participants ($N = 346$; 159 female, 187 male; mean age = 36; age range: 19–73) engaged in the same state-similarity rating task as in Study 3. However, they rated the pairwise similarities between a subset of just 10 states, and for just the close and far targets. This yielded a total of 90 trials. To maximize the expected effect size, the selected states were those that showed the greatest close-far difference in Study 3. Participants also rated both the close and far targets on similarity, familiarity, and closeness.

As in Study 3, responses were analyzed using linear mixed effects. Random effects included intercepts for participant and state pair, and slopes for target within each of these intercepts. The fixed effect of target was represented by a binary factor distinguishing the close and far targets. In addition, we conducted a second version of this analysis using a continuous social distance composite consisting of participants' ratings of the similarity, familiarity, and closeness (mean $r = 0.89$) of each target person averaged together.

### Data availability

Data and code from all four studies are freely available online on the Open Science Framework (https://osf.io/hp5wc/).

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

## Acknowledgements

The authors would like to thank Zidong Zhao, Meghan Meyer, Mai Nguyen, Aaron Kurosu, Adam Lerner, Tony Phan, and Ken Norman for their assistance. M.A.T. was supported by The Sackler Scholar Programme in Psychobiology. This work was supported by NIMH grant R01MH114904 to D.I.T.

## Author contributions

Conceptualization: M.A.T. and D.I.T. Software: J.N.M. and M.E.W. Formal analysis: M.A.T. and M.E.W. Investigation: M.E.W. Data curation: M.A.T and M.E.W. Writing—original draft preparation: M.A.T. Writing—review and editing: all authors. Visualization: M.A.T. Supervision: D.I.T.

## Additional information

**Competing interests:** The authors declare no competing interests.

