## [Peer Review File · Nature Communications]

Reviewers' comments:

Reviewer #1 (Remarks to the Author):

Review of "Representations of others' mental states grow less distinct with social distance" By Mark A. Thornton and colleagues.

Here the authors argue to test whether subjective and neural representations are more distinct for psychologically close versus distant individuals. This is done in the context of two well-powered fMRI experiments and two behavioural studies. I think the authors are asking a very interesting and psychologically important question and the RSA provides an interesting way for looking into the differences on how we represent others' mental states. fMRI imaging and analysis seems to be well done, and the sample size is decent. I however do not think the data supports the authors' claims, and I have some concerns regarding the interpretation of the neuroimaging results:

1. My main concern relates to the interpretation of the findings. Study 1 shows that mental state representations are different for self vs. other individuals. This is essentially a comparison between processing information about oneself versus others, which is a very different issue than social distance between two external agents, such as e.g. one's sister versus a stranger, as implied in the Introduction. The self and its processing simply differs radically from numerous other external and internal classes of stimuli and concepts (see e.g. oldish but still relevant review by Northoff et al 2006 NeuroImage), and the difference between close versus distant others is not tested in the study.

2. Study 2 introduces the comparison between self, close and distant others. The results show that self-referential processing differs from processing of i) close and ii) distant others. These are, again, a comparisons between processing information about oneself versus others. However, the conceptually critical difference between distant versus close others, is not significant. It is thus not too surprising that the RSA localizations in Experiments 1-2 map to those seen in studies tapping self-referential processing in univariate neuroimaging studies. Thus, the fMRI data do not support differential representation of mental states as a function of social distance, as long as "self" is not considered as "zero social distance" which would sound quite odd.

3. The critical comparison (close vs. distant others) is significant only behavioural Experiment 4, but again not in behavioural Experiment 3. Yet, the self-versus-distant-others difference is replicated in Experiment 3. Thus, I think the authors have strong evidence for arguing that we represent our own mental states differently than others' mental states, but only very limited evidence for the argument that mental state attributions for others become more diffuse when they are socially more distant.

4. In sum, the authors do not have solid data supporting their main argument, e.g. as they write in the abstract that "In two neuroimaging studies, representational similarity analyses demonstrated that neural representations of mental states become less distinct with increasing social distance". This is simply not backed up with the data. Just discussing the results in terms of differential representation of one's own versus others' mental states would be perfectly in line with the data, but I understand this may not sound as exciting and it lacks the novelty value of the current argument.

5. For the fMRI studies, it would also be interesting to see just old-fashioned t-maps between self and other referential processing – I reckon the RSA measures a different thing, but it would be imperative to test if there would just be a magnitude difference in the BOLD responses and if this maps to the same areas as the RSA.

6. The experiments are well powered, with +30 fMRI subject and +300 subjects in the behavioural Experiment 4. Despite this, the magnitude of the effects in the behavioural experiments seem very

modest. There is of course nothing wrong in this per se, but this should be clarified in the Discussion which makes quite strong claims regarding the "clarity" with which we represent others' minds – at least Figure 3C-D suggests that there is just a relatively small bump in state similarity as a function of social distance.

7. Based on above, the Discussion is stretching the data too far, with far-reaching claims regarding out-group biases, distinctiveness of mental state representations in socially distant individuals and so forth. Relatedly, the discussion is quite unspecific with respect to the actual neural mechanisms underlying the state representations.

8. When reading the discussion I actually started to wonder what were the neuroimaging studies strictly required for? The main argument of the paper is in the distinctiveness of the mental state representations, and this is conveniently addressed by the self-reports. Now, it is of course interesting to see the effects in the brain too, but the brain-level data do not really provide a mechanistic explanation regarding how the mental states are represented or how the representations are warped depending on who is the target of mentalization. The authors should make a much more convincing case regarding this part of the data so that it does not just stand out as a methodologically advanced gimmick.

9. The description of the MDS in the Methods section is hard to follow – maybe it's just me but it would be useful to streamline this part .

In sum, this is a potentially interesting dataset but I would advise against publishing the manuscript in its present form due to aforementioned reasons. I think the authors have two options for re-working the manuscript: 1) If increasing sample size for Studies 2-3 would show the expected close-versus-distant difference, the manuscript would be well in line with the current main argument. Alternatively, the authors could frame their findings clearly along the observed differences between self-versus-other-referential processing. This would of course change the paper's main point completely, but again it would be better in line with the data.

Reviewer #2 (Remarks to the Author):

The paper presents two imaging and two behavioral studies that demonstrate that as social distance to a target person increases, his/her mental states are represented in a less differentiated manner. The authors rely on a visual metaphor to derive this prediction in the introduction (spatially distal targets are seen with less clarity/differentiation), but in the general discussion advance two possible explanations: People know less well distal targets, and people care less about them.

The results seem clear, the methodology sound and the conclusion potentially important. I would like the authors to explain more how the mental states were selected. Are these primarily minute differences between mental states (i.e., nuances) that are "smoothened" in perceiving distal targets? Or is it the case that even major distinctions (e.g., friendly vs. hostile; happy vs. angry; worried vs. relaxed) that are undermined?

Both the visual analogy and Construal Level Theory would predict that major distinctions might be amplified with distance. In the visual domain, for example, distance facilitates perception of visual gestalt, and as a result might increase perceived difference between two figures that have different gestalts yet similar details.

Related to the previous question, is there evidence of reduced dimensionality in perception of distal targets?

Reviewer #3 (Remarks to the Author):

The authors developed a fascinating hypothesis, that people's neural representations of others' mental states become less distinct with increasing social distance, and tested it with a series of four thoughtful and carefully analyzed experiments. Three of these experiments included a "close other" and a "far other" condition, critical to distinguish their "social distance hypothesis" from an alternative hypothesis (the "self-other hypothesis") suggesting that representations of mental states are more distinct for the self, and less distinct for others regardless of their social distance.

In two of these three experiments, no difference was observed between the "close other" and the "far other". This lack of a significant difference could not be just attributed to large error bars: the difference between the means is very small (Figure 4B,C). In the remaining experiment, which included only behavioral data, a significant difference between "close other" and "far other" was observed. The authors conclude that the data support the "social distance" hypothesis.

However, the majority of experiments including a "close other" and a "far other" found no significant difference. It is possible that Study 4 yielded significant results because it used a larger group of participants (N=346, as compared to the N =46 for Study 3). However, it appears that the differences between Study 4 and Study 3 are not driven by a reduction in the variance in Study 4, but largely by a greater mean difference in Study 4 as compared to Study 3. Larger samples are expected to affect the variances, but not the means. The larger sample size in Study 4, therefore, does not explain the difference between the results in Study 3 and Study 4.

In light of these observations, my recommendations are:

- to replicate Study 4, in order to show that the result was not accidental, and that with a sufficiently large sample size, the effect of social distance is robust

- to tone down the claims that the fMRI data support the "social distance hypothesis" (for example in the discussion section the authors write: "Two fMRI and two behavioral experiments support the hypothesis that social distance diminishes the clarity with which people represent others' mental states") since among those four experiments only a behavioral experiment supports the "social distance hypothesis" as opposed to the "self-other hypothesis".

Reviewer #1:

1. *My main concern relates to the interpretation of the findings. Study 1 shows that mental state representations are different for self vs. other individuals. This is essentially a comparison between processing information about oneself versus others, which is a very different issue than social distance between two external agents, such as e.g. one's sister versus a stranger, as implied in the Introduction. The self and its processing simply differs radically from numerous other external and internal classes of stimuli and concepts (see e.g. oldish but still relevant review by Northoff et al 2006 NeuroImage), and the difference between close versus distant others is not tested in the study.*

Study 2 introduces the comparison between self, close and distant others. The results show that self-referential processing differs from processing of i) close and ii) distant others. These are, again, a comparisons between processing information about oneself versus others. However, the conceptually critical difference between distant versus close others, is not significant. It is thus not too surprising that the RSA localizations in Experiments 1-2 map to those seen in studies tapping self-referential processing in univariate neuroimaging studies. Thus, the fMRI data do not support differential representation of mental states as a function of social distance, as long as "self" is not considered as "zero social distance" which would sound quite odd.

The critical comparison (close vs. distant others) is significant only behavioural Experiment 4, but again not in behavioural Experiment 3. Yet, the self-versus-distant-others difference is replicated in Experiment 3. Thus, I think the authors have strong evidence for arguing that we represent our own mental states differently than others' mental states, but only very limited evidence for the argument that mental state attributions for others become more diffuse when they are socially more distant.

In sum, the authors do not have solid data supporting their main argument, e.g. as they write in the abstract that "In two neuroimaging studies, representational similarity analyses demonstrated that neural representations of mental states become less distinct with increasing social distance". This is simply not backed up with the data. Just discussing the results in terms of differential representation of one's own versus others' mental states would be perfectly in line with the data, but I understand this may not sound as exciting and it lacks the novelty value of the current argument.

We agree with this concern raised by Reviewer #1 and Reviewer #3. In the previous submission of this research, we focused primarily on the effect of social distance in our results. We were intrigued that the granularity of the close target, always fell in the middle of that for the self and the far target. However, the reviewers rightly point out that the differences between the close and far target,

while consistent, were small and sometimes non-significant; instead, the most robust and consistent difference was between the self and the two targets. Although there is some evidence, particularly from Study 4, that this effect might extend to social distance more generally, a self-other account is a better supported interpretation, particularly with respect to the neural representation of mental states. Thus, while we originally conducted these studies to investigate the effect of social distance on the granularity of mental state representation, the results support a simpler account: people represent their own states much more richly than they do others. As such, we have reframed the manuscript accordingly. We now focus on greater granularity with which people represent their states, relative to the states of others. Although the interpretation of the results has changed, we believe that thinking about differences in the granularity of mental state representation between self and other is as interesting and novel as the social distance interpretation we initially submitted, and even more empirically sound. In the discussion, we retain some mention of the possible generalization to social distance (p. 14). However, this portion of the discussion is now much-reduced in scope and tempered in strength.

2. *For the fMRI studies, it would also be interesting to see just old-fashioned t-maps between self and other referential processing – I reckon the RSA measures a different thing, but it would be imperative to test if there would just be a magnitude difference in the BOLD responses and if this maps to the same areas as the RSA.*

There is indeed a well-established literature on the effects of thinking about the self (versus others) on univariate brain activity. The most robust finding across such studies is that self-reflection elicits greater activity than thinking about others in ventral medial prefrontal cortex. Indeed, we replicate these findings in our data, showing greater activity in the MPFC for the self than for the other two targets. As the reviewer suggests, we now include a supplemental figure reflecting the size of this effect across the cortex (p. 36).

However, as the reviewer suggests, the neural patterns we analyze in the RSAs do indeed measure a “different thing.” We use (Pearson) correlations to measure the similarity between brain activity patterns, thus effectively z-scoring each pattern. As such, mean univariate activity differences between conditions – whether states or targets – cannot contribute to pattern similarity in a given searchlight volume or ROI. Thus, the greater activity observed for self versus others in the univariate contrasts cannot explain the RSA results. Naturally, the multivariate patterns we analyze are composed of univariate activity differences in individual voxels, but this is necessarily true of fMRI pattern analyses in general.

It is also worth pointing out that the behavioral data in Studies 3 and 4 are consistent with the multivariate interpretation of the data. Observing that MPFC is more active for self versus other does not imply that people should judge their own mental states as more (or less) distinct than others’. Only when the

multivariate patterns are analyzed does this prediction about behavior follow from the imaging data.

3. *The experiments are well powered, with +30 fMRI subject and +300 subjects in the behavioural Experiment 4. Despite this, the magnitude of the effects in the behavioural experiments seem very modest. There is of course nothing wrong in this per se, but this should be clarified in the Discussion which makes quite strong claims regarding the “clarity” with which we represent others’ minds – at least Figure 3C-D suggests that there is just a relatively small bump in state similarity as a function of social distance.*

It is true that the effects in the behavioral studies are relatively modest in size. In Study 4 this can be readily attributed to our focus on the elusive distinction between close and far others. As multiple reviewers noted, this effect was not observed at a statistically significant level in any of the three preceding studies. The particularly large sample in Study 4 was meant to provide sufficient power to detect this smaller effect. In this respect it appears to have succeeded, in the sense that mean state similarity ratings were different between close and other in the predicted direction. However, the overall size of this effect remains small in absolute or standardized terms, as one would expect.

There are many reasons why the magnitudes of neural and behavioral effects might diverge. These include psychologically interesting possibilities, such as the possibility that participants might feel uncomfortable reporting that their own states are more distinct than those of others, leading to a desirability bias. However, there are also many differences in methodology which could explain this apparent difference in effect size. For example, given the much greater amount of data collected per participant in the imaging experiments, the differences might be attributed to differential measurement error.

In any case, as noted in #1 above, we have tempered our claims considerably about the effect of social distance as such (p. 14). Instead, we now focus on the more statistically reliable difference between self and other, which we observe across both brain and behavior in Studies 1-3.

4. *Based on above, the Discussion is stretching the data too far, with far-reaching claims regarding out-group biases, distinctiveness of mental state representations in socially distant individuals and so forth. Relatedly, the discussion is quite unspecific with respect to the actual neural mechanisms underlying the state representations.*

As discussed in #1 and #3 above, we have substantially tempered our claims regarding the effects of social distance per se, particularly as they relate to any difference in the clarity of mental state representation between close and far others. We retain references to relevant effects such as the outgroup bias, as we believe these are useful for thinking about analog of the present findings.

However, we have added an extended, explicit caveat regarding the generalizability of the results of Study 4 (p. 14).

5. *When reading the discussion I actually started to wonder what were the neuroimaging studies strictly required for? The main argument of the paper is in the distinctiveness of the mental state representations, and this is conveniently addressed by the self-reports. Now, it is of course interesting to see the effects in the brain too, but the brain-level data do not really provide a mechanistic explanation regarding how the mental states are represented or how the representations are warped depending on who is the target of mentalization. The authors should make a much more convincing case regarding this part of the data so that it does not just stand out as a methodologically advanced gimmick.*

The use of neuroimaging in the present study provides three primary advantages. First, it is a form of convergent evidence. In general, we believe that any conclusion is strengthened by showing that multiple measures and paradigms point to the same conclusions.

Second, brain activity is a potent implicit measure. Although people can be trained via neurofeedback to control their brain activity patterns, there is no evidence that they can do so spontaneously, particularly with the degree of subtlety that would be necessary to interfere with the analyses of the present data. Without an extensive neuroscience education – which our participants did not generally have – we do not believe that people could subvert their natural responses in the present paradigm (other than by simply not engaging properly in the task).

We expected a strong potential for response bias in the explicit measures in the behavioral studies. Indeed, such biases are perhaps why the neural results are more robust than the behavioral results, as noted in #3 above. There is something profoundly strange about claiming that one's own mental states are more distinctive than those of another person. If the behavioral task had been set up to task the same question sequentially for each target (e.g., “how similar are x and y for you?” followed immediately by “how similar are x and y for [close target]”?) then we doubt that we would have observed any differences at all, due to consistency bias. The use of an implicit measure thus helps to compensate for potential biases in self-report. This ensures greater confidence that the results cannot be easily attributed to factors such as social desirability.

Finally, fMRI provides information about the localization of effects within the brain. This information is useful in two respects. First, it reveals more about the neural machinery of mental state representation. Previous studies have consistently indicated that a widely distributed set of brain regions are involved in representing others' thoughts and feelings. However, it is less clear whether these representations correspond to unchanging concepts or are malleable to change based on context. Localizing context-sensitive regions can start to reveal which portions of the social brain are repositories for knowledge, and which portions

flexibly draw up on and modify that knowledge to make inferences. The second way in which localization is useful is that it provides a reasonableness check on the interpretation of the behavioral effects. Although reverse-inference can be a fraught procedure, it can still be informative at times. For instance, in the present case we observed that changes in the distinctiveness of mental state representations are largely confined to regions which have been previously implicated in mental state representation in general. If this were not the case it would seriously qualify the results of the present investigation by undermining the claim that mental state representations were actually changing in place as a function of target, rather than being amended by some other neural system.

6. *The description of the MDS in the Methods section is hard to follow – maybe it’s just me but it would be useful to streamline this part.*

We have attempted to clarify the MDS procedure in the methods section (p. 33) and the figure caption (p. 11).

Reviewer #2:

7. *The results seem clear, the methodology sound and the conclusion potentially important. I would like the authors to explain more how the mental states were selected. Are these primarily minute differences between mental states (i.e., nuances) that are "smoothed" in perceiving distal targets? Or is it the case that even major distinctions (e.g., friendly vs. hostile; happy vs. angry; worried vs. relaxed) that are undermined? Both the visual analogy and Construal Level Theory would predict that major distinctions might be amplified with distance. In the visual domain, for example, distance facilitates perception of visual gestalt, and as a result might increase perceived difference between two figures that have different gestalts yet similar details. Related to the previous question, is there evidence of reduced dimensionality in perception of distal targets?*

We now provide an extended description of the state selection process (p. 31-32). We also discuss which particular states appear to show the greatest reduction in clarity as a function of the self-other distinction (p. 15).

We do not observe any increase in the importance of broad psychological dimensions as a function of increasing distance. Nor do we observe any consistent reduction in the dimensionality of the representational space across targets either. In Study 1 we observe that one dimension was more important for the self than for the far target, but this effect did not replicate in Study 2. In general, the “geometry” of mental state representation appears relatively robust across targets. That is, the size of the representational space may change, but the shape of that space – in terms of which dimensions are important in explaining neural similarity – seems quite similar regardless of whether one is thinking about one’s own states or those of others. We have added a section on dimensional representational similarity analysis to report these findings and how they were obtained (p. 34-35).

Reviewer #3:

8. *My recommendations are to replicate Study 4, in order to show that the result was not accidental, and that with a sufficiently large sample size, the effect of social distance is robust [and] to tone down the claims that the fMRI data support the “social distance hypothesis” or example in the discussion section the authors write: “Two fMRI and two behavioral experiments support the hypothesis that social distance diminishes the clarity with which people represent others’ mental states”) since among those four experiment only a behavioral experiment supports the “social distance hypothesis” as opposed to the “self-other hypothesis”.*

As described in the cover letter and point #1 above, we have adopted the second of these two recommendations, and have re-framed the present findings in terms of the more robust self-other distinction. We retain discussion of the more general social distance hypothesis (p. 14), but focus primarily on the greater granularity with which people seem to represent their own minds, relative to those of both socially close and socially distant others.

REVIEWERS' COMMENTS:

Reviewer #3 (Remarks to the Author):

I appreciate the changes made by the authors - now the interpretation of the findings matches more closely the empirical results.

Given the new framing of the results, it becomes even more important to clarify the novelty of the findings with respect to the univariate literature on self vs others. While it is true that stronger univariate responses for the self in prefrontal cortex do not necessarily imply more distinct response patterns, stronger univariate responses have been widely interpreted in the field as evidence for greater engagement/ the encoding of more information. For example, stronger responses to familiar faces in the posterior cingulate have been interpreted as evidence that this region encodes knowledge about familiar people (Gobbini and Haxby 2007). It would be appropriate to include a discussion of the univariate literature on prefrontal cortex responses to self vs others, and to mention possible alternative explanations for why PFC responses would be greater for the self (if not because more information is represented about the self). Laying out this hypothesis space would help clarify which hypotheses can be ruled out thanks to the present findings.

Reviewer #4 (Remarks to the Author):

I was not part of the initial review and so I may have missed some of the changes from first submission to the present version. This paper reports on an interesting set of studies examine whether the neural patterns associated with different mental states are more or less distinct for self and others. The paper is well-written, short (in a good way) and the analysis are clear and well-explained the methods and supplemental sections. The authors are well-experienced at this kind of research and, in my view, the appropriateness and validity of the statistical analysis are top notch. Either I'm getting soft in my old age or the authors did a good job of making the analysis and results clear such that I find myself having surprisingly few criticisms or comments or issues with the methods, procedure and analysis. Nevertheless, here's two comments:

(1) My response to the manuscript is very much in line with that of Reviewer 1 and I found the authors did a good job responding to their comments and questions. That said, I'd like to press the issue of how this differs from the many univariate studies that have looked at how the MPFC responds to self vs. others. Across multiple studies and domains it's often been found that the MPFC (we can quibble about precise location) is more active when thinking about the self vs. others (and sometimes even vs. friends). I fully agree with the authors that RSA does not measure the same thing and thus univariate effects do not account for the finding. But I don't see that as the point. Rather, for me the issue isn't that the analysis might be confounded by univariate effects, it's whether the finding represents a significant change in how we've been interpreting the univariate findings all along. For quite a while now, we've known that the MPFC does "something" more for self than others. Although inferences based on activation can be a bit fuzzy, a common view has been that the MPFC is more involved/has different knowledge of/has more access to representations of one's own self than that of others. Specifying what that something is warrants reporting, and the present findings certainly do just that, but in my view, this largely adds to -and could be incorporated into- the various interpretations of univariate self>other findings floating around.

(2) Is there a way, with the present data, of showing that this isn't simply an extension of the self being an overlearned and highly accessible cognitive structure for people? Something we are intimately familiar with? And would that change your interpretation discussion of the finding?

Localizing the effect to the "social brain" is interesting, but is the underlying phenomena somehow different than research showing that experts have a finer understanding of the things they are experts at? Bird watchers and car people have more distinct representations of birds and cars than I do. Japanese speakers have more difficulty distinguishing ra and la than English speakers, etc. I don't think that would necessarily diminish how interesting this study is, but it does put it in a different context (i.e., an extension of a general phenomenon related to familiarity and expertise).

Reviewer #3:

1. *Given the new framing of the results, it becomes even more important to clarify the novelty of the findings with respect to the univariate literature on self vs others. While it is true that stronger univariate responses for the self in prefrontal cortex do not necessarily imply more distinct response patterns, stronger univariate responses have been widely interpreted in the field as evidence for greater engagement/ the encoding of more information. For example, stronger responses to familiar faces in the posterior cingulate have been interpreted as evidence that this region encodes knowledge about familiar people (Gobbini and Haxby 2007). It would be appropriate to include a discussion of the univariate literature on prefrontal cortex responses to self vs others, and to mention possible alternative explanations for why PFC responses would be greater for the self (if not because more information is represented about the self). Laying out this hypothesis space would help clarify which hypotheses can be ruled out thanks to the present findings.*

We agree with the reviewer that it is useful to consider the present results in light of the established findings on self-other activity differences in the brain. As Supplementary Figure 1 shows, we replicate the standard self-reference effect at the univariate level in both Studies 1 and 2. Specifically, we observe much greater activity in a central portion of medial prefrontal cortex for the self condition than the other conditions. We also observe equally robust univariate differences in the opposite direction across much of the social brain network. The precuneus, for instance, is much less active for self than other in both fMRI studies. However, despite these opposite univariate effects, we observe a parallel pattern similarity effects in MPFC and precuneus in Study 2. That is, both of these regions manifest more distinctive patterns for one's own mental states than for the states of another person. This dissociation – and indeed reversal – between univariate and multivariate effects may qualify previous interpretations of greater activity as being a unique signature of increased self-referential processing. Instead, both increased and decreased univariate activity can lead to representing one's own states with greater granularity than those of others. This finding casts previous univariate findings in a new light and illustrates the novel implications of examining multivoxel patterns of brain activity. We now discuss the relation between the univariate and multivariate effects in the discussion p. 12.

Reviewer #4:

2. *My response to the manuscript is very much in line with that of Reviewer 1 and I found the authors did a good job responding to their comments and questions. That said, I'd like to press the issue of how this differs from the many univariate studies that have looked at how the MPFC responds to self vs. others. Across multiple studies and domains it's often been found that the MPFC (we can quibble about precise location) is more active when thinking about the self vs. others (and sometimes even vs. friends). I fully agree with the authors that RSA*

does not measure the same thing and thus univariate effects do not account for the finding. But I don't see that as the point. Rather, for me the issue isn't that the analysis might be confounded by univariate effects, it's whether the finding represents a significant change in how we've been interpreting the univariate findings all along. For quite a while now, we've known that the MPFC does "something" more for self than others. Although inferences based on activation can be a bit fuzzy, a common view has been that the MPFC is more involved/has different knowledge of/has more access to representations of one's own self than that of others. Specifying what that something is warrants reporting, and the present findings certainly do just that, but in my view, this largely adds to -and could be incorporated into- the various interpretations of univariate self>other findings floating around.

We agree with this reviewer that it is important to contextualize the present results in terms of the previous findings on univariate effects of self-referential thought. We have added a general discussion of how the results data replicate those previous findings – particularly the greater MPFC activity for self vs. other – at the univariate level. We also point out an interesting discrepancy the univariate and multivariate results. Specifically, we see regions such as the precuneus activating in the opposite direction from MPFC in the univariate contrast, but these regions show the same response in the multivariate analyses. Specifically, we observe in Study 2 that both regions show greater neural pattern distinctiveness for self versus other. In other words, opposite univariate activity differences correspond to similar multivariate representational differences. This discrepancy may indeed call for a significant reinterpretation of the previous univariate findings. Although the increased MPFC activity involved in self-referential thought still seems like a valid signifier of increased processing of self vs. other, the present findings suggest that increased activity is a not a unique signature of this enhanced self-processing. In other brain regions, enhanced self-referential thought may instead be reflected in decreased levels of univariate activity, a possibility largely overlooked in previous research. Thus the present findings do not undermine the conclusions drawn from past research regarding the self and MPFC, but they suggest that the methods available to and default assumptions made in those investigations may have led to them overlooking other valuable conclusions. We discuss these issues on p. 12.

- 3. Is there a way, with the present data, of showing that this isn't simply an extension of the self being an overlearned and highly accessible cognitive structure for people? Something we are intimately familiar with? And would that change your interpretation discussion of the finding? Localizing the effect to the "social brain" is interesting, but is the underlying phenomena somehow different than research showing that experts have a finer understanding of the things they are experts at? Bird watchers and car people have more distinct representations of birds and cars than I do. Japanese speakers have more difficulty distinguishing ra and la than English speakers, etc. I don't think that would necessarily diminish*

how interesting this study is, but it does put it in a different context (i.e., an extension of a general phenomenon related to familiarity and expertise).

The reviewer raises an interesting question regarding the source of the greater granularity of self-representation that we observe. On one hand, it might be that unique aspects of experience lead to the greater distinctiveness with which we represent our own minds. For example, we have access to data channels such as introspection for our selves but not for others, and this might lead to our tendency to make finer distinctions about our own states. On the other hand, the self might be special primarily by virtue of quantity, rather than quality. That is, by virtually every definition, everyone has far more experience with their own mental states than with the states of anyone else. This sheer quantity of experience may lead to our tendency to represent the self with greater granularity. Either of these factors could in principle lead to the enhanced self-referential processing on its own. The unique nature of the self could also be overdetermined, with both of these factors contributing to it in unique ways.

Experimentally testing the contribution of introspective access and similar features to self-representation would be a challenging undertaking. We are not presently aware of an approach that could specifically manipulate introspective access. Manipulating the insight one has into others' minds – effectively mimicking introspective access – may prove a more effective approach to studying this side of the question than manipulating people's own introspection. However, it might be possible to find clinical or neuropsychological populations who have lost insight into themselves as a function of their conditions, and compare the self-representations of these individuals to neurotypical comparisons

The question of overlearning the self is perhaps more easily amenable to experimental investigation. Here, for instance, we investigated not just self-other, but continuous social distance in Studies 2-4. Insofar as greater quantity of experience with a person leads us to represent their states with greater distinctiveness, we should have observed that in the comparison of close and far others in those studies. We did observe weak evidence in support of this hypothesis – directionally in Studies 2-3, and significantly in Study 4. However, the size of this effect, if it exists, appears to be much weaker than the self-other discrepancy. Of course, the enormous quantity of experience with the self may be hard to compare with the quantity of experience one has with even a close friend, so this is not conclusive. Moreover, the social distance manipulation was multifaceted, including similarity and liking as well as familiarity, so it we could not uniquely attribute these effects to experience volume per se.

We discuss this issue on p. 10 and suggest that future investigations target the effects of introspection and familiarity on state distinctiveness.